# Noradrenergic Pathways Involved in Micturition in an Animal Model of Hydrocephalus—Implications for Urinary Dysfunction

**DOI:** 10.3390/biomedicines12010215

**Published:** 2024-01-18

**Authors:** Marta Louçano, Ana Coelho, Sílvia Sousa Chambel, Cristina Prudêncio, Célia Duarte Cruz, Isaura Tavares

**Affiliations:** 1Unit of Experimental Biology, Department of Biomedicine, Faculty of Medicine, University of Porto, 4200-319 Porto, Portugal; lpm@ess.ipp.pt (M.L.);; 2IBMC-Institute of Molecular and Cell Biology, University of Porto, 4200-135 Porto, Portugal; 3I3S-Institute of Investigation and Innovation in Health, University of Porto, 4200-135 Porto, Portugal; 4Chemical and Biomolecule Sciences, School of Health, Polytechnic of Porto, 4200-072 Porto, Portugal; 5Center for Translational Health and Medical Biotechnology Research (TBIO), Polytechnic of Porto, 4200-072 Porto, Portugal

**Keywords:** hydrocephalus, periaqueductal gray, locus coeruleus, urinary dysfunction, noradrenaline, Onuf’s nucleus

## Abstract

Hydrocephalus is characterized by enlargement of the cerebral ventricles, accompanied by distortion of the periventricular tissue. Patients with hydrocephalus usually experience urinary impairments. Although the underlying etiology is not fully described, the effects of hydrocephalus in the neuronal network responsible for the control of urination, which involves periventricular areas, including the periaqueductal gray (PAG) and the noradrenergic locus coeruleus (LC). In this study, we aimed to investigate the mechanisms behind urinary dysfunction in rats with kaolin-induced hydrocephalus. For that purpose, we used a validated model of hydrocephalus—the rat injected with kaolin in the cisterna magna—also presents urinary impairments in order to investigate the putative involvement of noradrenergic control from the brain to the spinal cord Onuf’s nucleus, a key area in the motor control of micturition. We first evaluated bladder contraction capacity using cystometry. Since our previous characterization of the LC in hydrocephalic animals showed increased levels of noradrenaline, we then evaluated the noradrenergic innervation of the spinal cord’s Onuf’s nucleus by measuring levels of dopamine β-hydroxylase (DBH). We also evaluated the expression of the c-Fos protooncogene, the most widely used marker of neuronal activation, in the ventrolateral PAG (vlPAG), an area that plays a major role in the control of urination by its indirect control of the LC via pontine micturition center. Hydrocephalic rats showed an increased frequency of bladder contractions and lower minimum pressure. These animals also presented increased DBH levels at the Onuf´s nucleus, along with decreased c-Fos expression in the vlPAG. The present findings suggest that impairments in urinary function during hydrocephalus may be due to alterations in descending noradrenergic modulation. We propose that the effects of hydrocephalus in the decrease of vlPAG neuronal activation lead to a decrease in the control over the LC. The increased availability of noradrenaline production at the LC probably causes an exaggerated micturition reflex due to the increased innervation of the Onuf´s nucleus, accounting for the urinary impairments detected in hydrocephalic animals. The results of the study provide new insights into the neuronal underlying mechanisms of urinary dysfunction in hydrocephalus. Further research is needed to fully evaluate the translational perspectives of the current findings.

## 1. Introduction

Several infectious, traumatic, or developmental pathologies may affect neuronal control of the Lower Urinary Tract (LUT), resulting in urinary impairments. Hydrocephalus is a disease that represents a major concern due to the major morphological and functional alterations caused by the impairment of the flow of the cerebrospinal fluid (CSF) and consequent enlargement of the cerebral ventricles [1,2]. The dilation of the ventricles induces damage to the adjacent tissues, at first, by stretching periventricular axons; with the progression of the disease, small blood vessels are compressed, leading to impairments in brain metabolism and neurochemical function [3,4]. Hydrocephalus is characterized by a triad of symptoms, including gait disturbance, cognitive impairments, and urinary incontinence [5,6]. LUT symptoms (LUT) are commonly detected in hydrocephalic patients and may be a significant cause of bother for both patients and caregivers. Patients report storage symptoms such as urinary frequency, urgency, urinary incontinence, and voiding symptoms, including retardation in initiating urination, prolonged/poor flow, the sensation of post-void residual volume (PVR), straining, and intermittency [7]. Urinary urgency and frequency are more common than urinary incontinence [7].

The bladder operates in a switch-like manner, alternating between storage and periodic voiding. The switch between the two phases is mediated by the spinobulbospinal voiding reflex. Voiding is under voluntary control and depends on learned behavior established during the maturation of the nervous system [8]. The periaqueductal grey (PAG) plays an important role in the neuronal control of the LUT. Information from the bladder reaches the PAG through the spinal cord, with a relay at the pontine micturition center (PMC) [9,10]. Stimulation of the PAG shows that different local areas are involved in micturition control, with a key role of the ventrolateral (vlPAG) [10,11,12]. The vlPAG has direct connections with the spinal cord and is the main column receiving afferents from the lumbosacral segments [10,11,12,13]. During the storage phase, the PAG is activated, and the PMC is inactive. When the bladder volume threshold is reached, the switch from storage to voiding is associated with an increase in the PAG activity, and the PMC is activated, inducing contraction of the detrusor synchronized with a relaxation of the external urethral sphincter (EUS), allowing for urine elimination [10,11]. As to the descending pathways, projections from the PMC neurons diverge to innervate the noradrenergic pontine locus coeruleus (LC) and reach the lumbosacral parasympathetic neurons of the Onuf´s nucleus [14]. Noradrenergic neurons from the LC innervate both the intermediate and ventral horn of the lumbosacral spinal cord, where pre-motor and motoneurons from the Onuf´s nucleus that control both the bladder and EUS are located. This pattern of innervation allows the synchronization of contraction and relaxation of the bladder and EUS, respectively, during voiding [15]. In addition to PMC projections, the LC also receives sensorial information from the bladder through the vlPAG [16,17]. As for the Onuf’s nucleus, it encompasses a small group of motoneurons that innervate the bladder and EUS, controlling the pelvic-perineal muscles and playing a crucial role in urinary continence [18]. The descending noradrenergic system from the LC regulates those neurons and, hence, is able to modulate the micturition reflex [19,20]. 

The exact mechanism of bladder dysfunction in hydrocephalic patients remains uncertain. Most hydrocephalic patients present storage symptoms, and some report urinary urgency and/or frequency without urinary incontinence, suggesting that urinary urgency/ frequency might precede urinary incontinence in hydrocephalus [7]. Urodynamic evaluation shows that in hydrocephalic patients, bladder dysfunction results from detrusor overactivity (DO), suggesting altered brain autonomic control in this disorder [5,7]. Indeed, a recent study reported small bladder capacity and increased post-void residual volumes (PVRs) in many patients [7]. 

Since knowledge about the mechanism behind urinary dysfunction during hydrocephalus is scarce and considering that two key areas involved in the neuronal control of micturition are affected in this condition, namely the PAG and the LC, our aim was to determine the micturition profile of hydrocephalic animals and evaluate the degree of neuronal activation at the PAG. We have recently demonstrated that the LC undergoes major alterations in hydrocephalic animals, with an increase in local levels of noradrenaline-synthetizing enzymes, which affect descending pain modulation through increases in noradrenergic innervation of the spinal dorsal horn [21]. We now continue the characterization of neurological dysfunctions in hydrocephalus. Due to these changes in noradrenaline synthesizing enzymes in the LC, and since the LC is a PAG relay station to the spinal cord, we also aim to evaluate the levels of the noradrenaline synthesizing enzyme dopamine-β-hydroxylase (DBH) in the Onuf´s nucleus of hydrocephalic animals. To answer those questions, we used a validated animal model, the rat injected with kaolin in the cisterna magna, used in our previous studies about noradrenergic pain modulation in hydrocephalic animals [2,22,23].

## 2. Materials and Methods

All procedures described above were approved by the Institutional Animal Care and Use Committee of the Faculty of Medicine at the University of Porto in Portugal. The experiments were conducted in accordance with the ethical guidelines for pain investigation and the European Community Council Directive (2010/63/EU). The Animal Ethical Committee of the Faculty of Medicine at the University of Porto and the Directorate-General of Food and Veterinary Medicine–Portuguese National Authority for Animal Health approved the experiments under license number 0421/000/000/2020. A total of 30 animals (Saline *n* = 13; Kaolin *n* = 17) were used, and all experiments were performed in pathogen-free male Wistar rats derived from the Faculty’s colony, housed in standard Plexiglass cages with free access to food and water. The colony room was maintained in a controlled environment (22 ± 2 °C; humidity, 55 ± 5%) on a standard 12/12 h light-dark cycle. The animals were weighed daily, and their condition was assessed by the veterinary staff to ensure their welfare. All experiments were conducted during the light phase. 

### 2.1. Surgical Induction of Hydrocephalus

Surgical induction of hydrocephalus was conducted in Wistar rats at the postnatal 21-day [24,25]. The animals were anesthetized by an injection into the abdomen (intraperitoneal injection). The injection was a mix of ketamine hydrochloride (0.06 g/kg; Imalgene™, Boehringer Ingelheim, Ingelheim am Rhein, Germany) and medetomidine (0.25 mg/kg; Medetor™, Virbac, Sintra, Portugal). After the anesthesia, the animals were placed on a stereotaxic frame (David Kopf Instruments; Tujunga, CA, USA) to reach the cisterna magna. During the surgical procedure, the anesthesia was maintained using a volatile anesthetic system (isoflurane (Isoflo, Abbott Animal Health, Madrid, Spain) 1% in 40% oxygen/air mixture) delivered at a constant rate. The region of the head and neck was shaved, and under a binocular loupe, the cervical muscles were carefully separated to expose the atlantooccipital membrane. A sterile Kaolin (Sigma-Aldrich, St. Louis, MO, USA) 20% suspension in a volume of 0.04 mL was slowly injected into the cisterna magna at a slow rate with a 27 G needle, as previously described [21,25]. The Kaolin injection was administered after the CSF aspiration to confirm access to intracisternal space. The needle was left in place for a few seconds after injection to prevent reflow and was then gently removed. Control rats underwent the same procedure as hydrocephalic animals but received an injection of 0.9% sterile saline of the same volume as kaolin suspension. Following the surgery, all animals were monitored daily to evaluate body weight, head circumference, gait, and general appearance for any signs of hydrocephalus development. Like human patients, rats with kaolin-induced hydrocephalus present gait disturbances [2,26,27]. A set of reflexes and motor tests were conducted to evaluate the motor function of the animals. The included placing/stepping reflexes, withdrawal reflex, toe spread reflex, observation of posture and ambulation abilities, and a variation of the tail suspension test [28,29]. The placing reflex was assessed by rubbing the rat’s feet against the table edge to evaluate the speed and accuracy of foot placement. The evaluation of withdrawal reflexes involved assessing the speed and force of hindlimb withdrawal when stimulated. The rat’s spontaneous activity was observed to assess posture and ambulation abilities. The tail suspension test was performed by lifting the rat’s tail to induce hindlimbs to lose contact with the ground, making the rat move forward [28,29]. 

### 2.2. Cystometric Analysis

Eight weeks after induction of hydrocephaly, cystometric analysis was performed in hydrocephalic and control animals. Anesthesia was induced by subcutaneous injection of urethane (1.2 g/kg; Sigma-Aldrich, St. Louis, MO, USA), and body temperature was maintained at 37 °C with a heating pad. A suprapubic midline laparotomy was used to expose the bladder, and a 21 G needle was inserted into the bladder dome connected to an infusion pump and to a pressure transducer. Animals were left untouched for about 15–30 min to allow bladder stabilization. After that, cystometry was performed with room temperature saline infused at a constant rate of 6 mL/h, and the intraluminal pressure was measured and recorded for 1 h [30]. Cystometrograms were analyzed, and frequency, amplitude, and peak and basal pressure of bladder reflex contractions were quantified using the LabScribe V2 software (iWorx, World Precision Instruments, Friedberg, Germany).

### 2.3. Vascular Perfusion and Material Processing for Immunohistochemical Analysis

Following the cystometries, the animals were given an overdose of sodium pentobarbital (65 mg/kg; Eutasil, MedVet; Bragança, Portugal) via i.p. injection to induce a profound anesthesia. The animals were then placed in a supine position, and the thorax and abdomen were opened to expose the heart and perform fixation of the tissues by perfusion. A catheter was inserted into the ascending aorta, and 200 mL of calcium-free Tyrode’s solution was delivered, followed by 1 L of a fixative solution composed of 4% paraformaldehyde in 0.1 M phosphate buffer, pH 7.2. After perfusion, the brain and spinal cord segments L4–S1 were removed and immersed in fixative for a post-fixation period of 4 h, followed by 30% sucrose in 0.1 M PB, pH 7.2 overnight at 4 °C. The material was cut in a cryostat at 20 µm in coronal orientation, collected in 5 sets, and stored in a cryoprotection solution at −20 °C. One set of the L6 sections was used to evaluate DBH expression at the Onuf´s nucleus.

### 2.4. Immunohistochemical Analysis of DBH and Fos Expression

The studies of the noradrenergic enervation of the Onuf’s nucleus were performed in spinal L6 sections in the same group of animals that underwent cystometric analysis. DBH immunostaining was elected to identify noradrenergic fibers as this is a specific noradrenergic marker widely used to map noradrenergic innervation of the central nervous system [31,32].

Sections were carefully washed with 0.1 M phosphate buffer solution (PBS) and treated with 1% hydrogen peroxidase to inhibit endogenous peroxidase. The sections were then incubated with blocking solution (10% normal horse serum (NHS) in 0.3% Triton-X 25% in PBS with 0.1 M glycine) before incubation with the primary antibody, a monoclonal anti-DBH antibody raised in mouse (Merck Milipore, Burlington, Massachusetts, EUA, catalogue No. MAB308), diluted at 1:5000 in 0.1 M PBS in 0.1 M PBS containing 0.3% Triton X-100 (PBS-T) and 2% of NHS, for 24 h at room temperature. After washing with PBS-T, sections were incubated for 1 h with a horse biotinylated anti-mouse serum (Vector Laboratories, Newark, California, USA, BA2000) diluted in PBS-T containing 2% NHS. Sections were washed again and incubated for 1 h in PBS-T containing avidin-biotin complex (1:200; ABC; Vector Laboratories, Burlingame, CA, USA). After washing in 0.1 M Tris-HCl, pH 7.6, bound peroxidase was revealed using 0.0125% 3,3′-diaminobenzidine tetrahydrochloride (DAB: Sigma Aldrich, St. Louis, MO, USA) and 0.025% H_2_O_2_ in the same buffer. The sections were mounted on gelatine-coated slides, cleared in xylol, and cover slipped with Eukitt (Sigma, St. Louis, MO, USA). Photomicrographs of spinal cord sections were taken using a light microscope (Axioskop 40 model, Zeiss^®^, Oberkochen, Germany) with a high-resolution digital camera (Leica EC3 model) maintaining the same exposure and light settings. DBH immunoreactivity was quantified on 5 non-contiguous spinal cord sections selected at random from each L6 segment. The images were analyzed using ImageJ V1.53k (National Institute of Health, Madison, WI, USA) to identify the percentage of DBH-positive pixels in the grey matter of the Onuf’s nucleus. The mean of background level was calculated for each section by analyzing a small area without visible DBH fibers. To set a threshold level for DBH-positive pixels, a value of 5 standard deviations above the mean background level was used [33]. The mean background level at the spinal dorsal horn was determined for each section using region of interest (ROI) analysis [33].

As to the studies of the expression of Fos, the protein synthesized after the activation of the c-fos proto-oncogene, we used a group of Wistar male rats weighing 285–300 g at the time of induction. These experiments were not performed on animals with hydrocephalus induction at p21 due to the friability of the brain tissue, which impaired histological sectioning. The immunohistochemical detection of Fos was elected because it is the most widely and reliably used functional anatomical marker of activated neurons in the central nervous system [34,35,36]. The induction of hydrocephaly was performed as above, but each animal received 0.05 mL of 20% Kaolin suspension (*n* = 8) or saline (*n* = 6), as described previously [21]. All the immunohistochemical procedures were similar to the DBH immunostaining but the blocking solution was 10% normal swine serum (NSS), the primary antibody was a polyclonal anti-Fos antibody raised in rabbit (Calbiochem, San Diego, CA, USA, Cat. No. PC38), diluted at 1:20,000 and containing 2% NSS, and the incubation lasted 48 h at 4 °C and the secondary antibody was a swine biotinylated anti-rabbit serum (Dako, Copenhagen, Denmark, EO353s), diluted in PBS-T containing 2% NSS. After mounting as above, the numbers of Fos-immunoreactive (IR) neurons in the PAG were counted using a Nikon^®^ light microscope. The cytoarchitectonic division of the PAG in columns was conducted following the Atlas of Paxinos and Watson (2004) [37].

### 2.5. Statistical Analysis

The statistical analysis of the results from the cystometric studies was conducted by using an unpaired *t*-test to compare the control and Kaolin groups. Similarly, the results of the immunohistochemical studies were analyzed using an unpaired *t*-test to compare the control and Kaolin groups. The computer program used for this analysis was GraphPad Prism 6. Statistical significance was considered for *p*-values ˂ 0.05.

## 3. Results

### 3.1. General Conditions of the Animals

In the days after being injected with Kaolin, animals exhibited signs of hydrocephalus, including reduced activity levels, coordination difficulties, abnormal posture, hind leg weakness, and nasal and/or orbital secretion. These findings align with previous studies [2,30]. Moreover, some hydrocephalic animals developed an enlarged, dome-shaped head within one week of Kaolin injection, and varying degrees of unsteady gait were also observed, consistent with other studies. [2] After the first week, the animals of the hydrocephalic group recovered and gained weight normally as control animals. Concerning the motor function of hydrocephalic animals, in general, they present some motor impairments since most of them present a modest or no reflex. Regarding motor assessment, they show some difficulty in walking when forced to walk without the hindlimbs in the tail suspension test and show less exploratory activity compared to control animals.

### 3.2. Cystometric Analysis

The results of the cystometric analysis are shown in Figure 1. Several cystometric parameters, such as frequency, peak and basal pressure, and amplitude of voiding bladder contractions, were analyzed. Hydrocephalic animals showed an increase in the number of bladder contractions per minute (*p* = 0.0125; Figure 1A,C) and an increase in basal pressure of voiding contractions (*p* = 0.0305; Figure 1B,E). No statistically significant differences were detected in what concerns amplitude (*p* = 0.1590; Figure 1D) and peak pressure (*p* = 0.2233; Figure 1F). 

### 3.3. Hydrocephalic Animals Presented Increases in the Expression of Noradrenaline Synthetizing Enzyme (Dopamine-β-Hydroxylase) in the Onuf’s Nucleus

The results of the evaluation of DBH expression by immunohistochemistry were performed in the group of animals submitted to cystometric analysis in spinal cord sections obtained at the L6 level, which are shown in Figure 2. The Onuf’s nucleus, a distinct group of motoneurons located in the ventral part of the anterior horn of the lumbosacral region of the spinal cord, is delimited in Figure 2A,B. Hydrocephalic animals showed a higher number of fibers immunoreactive to DBH in comparison with control animals (* *p* = 0.0496; Figure 2C). 

### 3.4. Hydrocephalic Animals Presented Decrease in Neuronal Activation of the vlPAG

The evaluation of Fos expression by immunohistochemistry was performed in the group of adult rats in sections containing the complete rostrocaudal extension of the PAG. The results are shown in Figure 3. A decrease in the total number of Fos-IR cells in the PAG of the animals with hydrocephalus (* *p* < 0.05; Figure 3C). After that global analysis of the PAG (Figure 3C), we then analyzed its different columns separately to evaluate if that reduction was mainly restricted to a specific column. We analyzed the dorsomedial periaqueductal gray (dmPAG; Figure 3D), the dorsolateral periaqueductal gray (dlPAG; Figure 3E), the lateral periaqueductal gray (lPAG; Figure 3F) and the ventrolateral periaqueductal gray (vlPAG; Figure 3G). The decrease in the number of Fos-IR cells only reached statistical significance in the vlPAG (*** *p* < 0.0001; Figure 3G).

## 4. Discussion

To the best of our knowledge, this is the first study to evaluate the noradrenergic pathways involved in urinary control during hydrocephalus using a validated model of the disease. In these animals, the cystometric analysis showed an increase in the frequency and basal pressure of voiding contractions, accompanied by decreased Fos expression at the PAG, namely at the vlPAG, and increased levels of expression of DBH, an enzyme involved in noradrenaline synthesis. The implications of these data for the pathophysiology of LUT dysfunction in hydrocephalic patients need to be evaluated.

Hydrocephalic animals presented an increase in the frequency and basal pressure of bladder contractions in comparison with control animals. Furthermore, these animals also showed increased levels of DBH, the enzyme involved in the synthesis of noradrenaline, at the Onuf’s nucleus, where the motoneurons that regulate bladder contraction are located. We previously reported an increase in noradrenergic innervation in hydrocephalic animals at the LC [21]. The spinal noradrenergic system has a modulatory role in the control of the micturition reflex, and it is likely that an association between altered micturition and changes in noradrenergic innervation at the Onuf´s nucleus occurs. Pharmacological studies have shown that in cases of established DO, the administration of an alpha1 receptor antagonist (doxazosin) supplemented with a decrease in the amplitude of bladder contractions [19] has been effective in suppressing the DO. We will evaluate the levels and function of α1 receptors in the spinal cord in hydrocephalic animals in order to better uncover the mechanisms involved in changes in micturition during hydrocephalus. 

The impact of hydrocephalus on the neural brain circuits involved in the control of micturition should also be considered. Descending noradrenaline projections that innervate the lumbosacral segments of the spinal cord are required for the coordination of bladder contractions and EUS relaxation during voiding [15,38]. We previously showed that the PAG is strongly affected by the dilation of the cerebral ventricles in the hydrocephalus model used in the present study [21]. Here, the PAG was analyzed to evaluate the expression of the c-Fos protooncogene, as this is an indicator of neuronal activity [39]. Kaolin-injected animals presented a decrease in c-Fos levels in the PAG, mainly at the vlPAG column, which indicates decreased activation of this nucleus. Due to the role of the PAG in the voiding reflex [9,19], decreased neuronal activation is likely to affect the neuronal circuity involved in the micturition control. It is well established that the PAG, and mainly the vlPAG column, has an essential role in the spinobulbospinal voiding reflex pathway, receiving and integrating/modulating information from the bladder and higher centers of the brain. The descending limb of the voiding reflex passes from the PAG and reaches the bladder through projections from the PMC, and the appropriate motor responses needed to voiding can be generated in the motor neurons of Onuf’s nucleus [8,11,12]. The PAG contains a dense population of gamma-aminobutyric-acid (GABA)-ergic interneurons that apply a tonic inhibitory influence in the voiding reflex during the storage phase [13,40], promoting the relaxation of the detrusor muscle and the contraction of the EUS [11,13]. Activation of the vlPAG GABAergic neurons delayed detrusor contractions and blocked voiding via the PMC [13,40]. It is possible that the decrease in neuronal activation detected in the vlPAG of hydrocephalic rats represents an impairment in local GABAergic inhibition.

Since our hydrocephalic animals presented an increase in the frequency of bladder contractions and increased basal pressure, together with lower neuronal activation of the vlPAG, we suggest that one possible mechanism involved in urinary dysfunction in this condition can be due to the decreased inhibition that may silence the voiding reflex, during the storage phase, from the vlPAG. Pharmacological blockade of inhibitory GABAergic synapses from the vlPAG leads to an increase of tonic bladder contractions, preventing efficient emptying of the bladder and consequent increase in basal pressure [13]. It would be interesting to evaluate GABA levels in the PAG of hydrocephalic animals to evaluate if the increased frequency of voiding contractions and of basal pressure is due to decreased GABAergic inhibition by the PAG. The PAG also plays a crucial role during the storage phase since it receives ascending information about the bladder’s filling and, as it approaches the threshold level, regulates both the LC and the PMC [17]. Activation of the LC promotes a change in attention focus from non-voiding to voiding behavior [9]. The LC functions in a synchronized way with the PMC to coordinate focus, behavior, and bladder contractions during voiding.

In summary, we propose that ventricular dilation during hydrocephalus induces a decrease in PAG activation, probably due to increased GABAergic inhibition, mainly at the vlPAG column, probably via a decrease of GABAergic inhibition, with an impact at the LC. Local increases in noradrenaline levels will affect descending noradrenergic input to the Onuff´s center, probably causing an exaggerated micturition reflex. The causality relation of the findings leading to the proposed mechanisms needs to be demonstrated, along with the translational perspectives of the current findings. One important aspect to be considered from a translational perspective is the fact that the current study was only performed in male animals as part of a project aiming to study the neuronal circuits affected by hydrocephalus [21]. Herein, studies in female animals are necessary to increase the translational perspectives of the present study. We propose to perform some studies using translational approaches to fully demonstrate the proposed mechanisms. Focusing on brain circuits, which are crucial in the control of micturition [5,41] but are understudied in comparison with peripheral causes, we propose to use neuroimaging techniques to study neuronal activation and metabolites in the brain of hydrocephalic animals [5,41]. In hydrocephalic patients, imaging techniques have shown that urinary dysfunction can be due to an exaggerated micturition reflex to a frontal hypoperfusion [7]. It would be interesting to use the same imaging approach in hydrocephalic animals to correlate the activity of the brain-bladder control network with the changes in the micturition responses. Since changes in the micturition reflex detected in hydrocephalic patients [6] have been related to reduced central inhibition [42], the neuroimaging and metabolite analysis in the brain of hydrocephalic animals should focus on GABAergic modulation. Another evaluation is more directed to noradrenergic control. In fact, we have previously shown that the increase in noradrenaline levels in the LC of hydrocephalic animals is associated with a local increase of 8-OHdG, an oxidative stress marker. Recently, the antioxidant agent Evadaravone has shown positive effects in the treatment of several diseases, including hydrocephalus [43]. Taking into account these beneficial effects of Evadaravone and the current results, it would be interesting to preventively treat hydrocephalic animals with that antioxidant agent to evaluate if there are improvements in urinary dysfunction, along with normalization of noradrenaline levels at the LC and Onuff´s nucleus. 

## Figures and Tables

**Figure 1 biomedicines-12-00215-f001:**
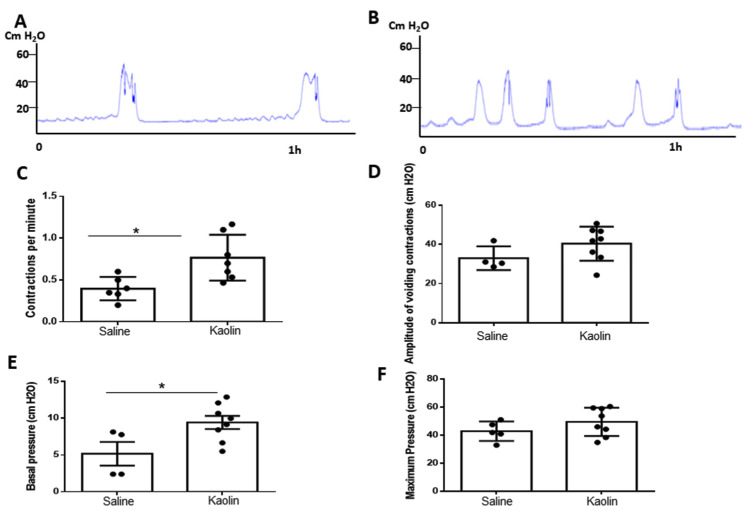
Cystometric evaluation performed on the animals 8 weeks after Saline injection (**A**) or Kaolin injection (**B**). Hydrocephalic animals show an increase in the frequency of bladder contractions per minute (**C**) (Saline *n* = 7; Kaolin *n* = 9) and an increase in basal pressure of voiding contractions (**E**) (Saline *n* = 6; Kaolin *n* = 9). No statistically significant differences were detected in Amplitude (**D**) (Saline *n* = 5; Kaolin *n* = 9) or Peak Pressure (**F**) (each dot represents one animal; Saline *n* = 5; Kaolin *n* = 9). Data in (**C**–**F**) are presented as means ± SEM. * *p* < 0.05.

**Figure 2 biomedicines-12-00215-f002:**
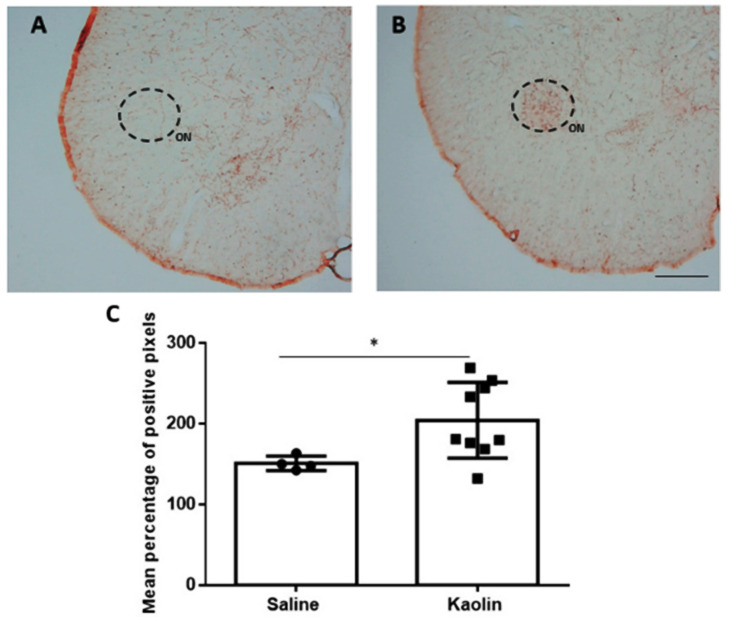
Expression of DBH at the Onuf’s Nucleus. Representative photomicrographs of DBH-immunolabelling in Saline- (**A**) and Kaolin-injected animals (**B**). Kaolin-injected animals presented increases in DBH expression (**C**) in the Onuf´s Nucleus. Each dot represents one animal Saline *n* = 4; Kaolin *n* = 9. Scale bar: 500µm (**A**,**B**) are at the same magnification. Data in (**C**) are presented as means ± SEM. * *p* < 0.05. ON: Onuf´s Nucleus highlighted in the dotted circles.

**Figure 3 biomedicines-12-00215-f003:**
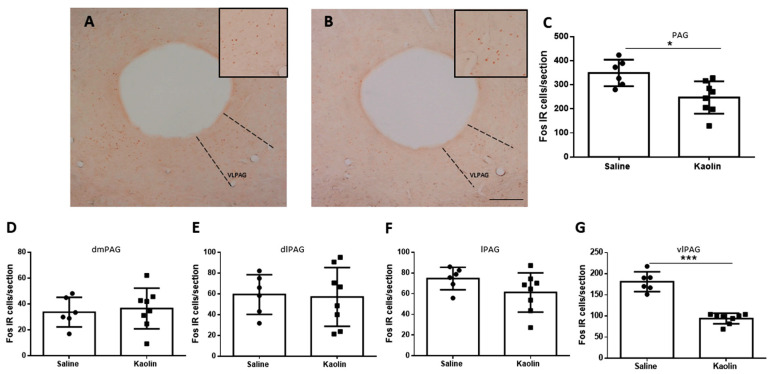
Neuronal activation of the PAG. Representative photomicrographs of Fos-IR neurons in the PAG after saline (**A**) and Kaolin (**B**) injections. Hydrocephalic animals show a decrease in c-Fos expression in the PAG (**C**), making this decrease more significant in the vlPAG column (**G**). No statistically significant differences were detected in the dmPAG (**D**), dlPAG (**E**), and lPAG (**F**) columns of the PAG. Saline *n* = 6; Kaolin *n* = 8. Scale bar: 1 mm (**A**,**B**) are at the same magnification. Data in (**C**–**G**) are presented as means ± SEM. * *p* < 0.05; *** *p* ˂ 0.001. PAG: periaqueductal gray; dmPAG: dorsomedial periaqueductal gray; dlPAG: dorsolateral periaqueductal gray; lPAG: lateral periaqueductal gray; vlPAG: ventrolateral periaqueductal gray, highlighted in the squares and dotted lines.

## Data Availability

All data presented in this study are available on request by contacting the corresponding author.

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
