# Peer review of "Noradrenergic Pathways Involved in Micturition in an Animal Model of Hydrocephalus—Implications for Urinary Dysfunction"

_biomedicines, 2024, doi:10.3390/biomedicines12010215_

Round 1

Reviewer 1 Report

Comments and Suggestions for Authors

The author creatively discusses the neural mechanism of urinary function impairment in patients with hydrocephalus, but the discussion of the mechanism can be further deepened. No matter the expression of DBH at the Onuf’s Nucleus or the decreasing in neuronal activation of the vlPAG in hydrocephalic animals, the author only made a symbolic observation and did not explore the mechanism. In general, the author needs to increase the necessary mechanism to increase the scientific significance of the article.

Author Response

Response to reviewers- Manuscript ID: biomedicines-2797923 “Noradrenergic pathways involved in micturition in an animal model of hydrocephalus – implications for urinary dysfunction”

Reviewer 1

The author creatively discusses the neural mechanism of urinary function impairment in patients with hydrocephalus, but the discussion of the mechanism can be further deepened. No matter the expression of DBH at the Onuf’s Nucleus or the decreasing in neuronal activation of the vlPAG in hydrocephalic animals, the author only made a symbolic observation and did not explore the mechanism. In general, the author needs to increase the necessary mechanism to increase the scientific significance of the article.

ANSWER: We agree with this remark and changed the manuscript accordingly to propose a mechanism based on the current and previous findings. We changed the abstract to summarize the mechanism proposed (please see lines 35-39) and the Discussion (please see the paragraph that starts in line 366).

Reviewer 2

For investigating the mechanisms underlying urinary dysfunction in a rat model with kaolin-induced hydrocephalus, the authors used cystometric analysis for evaluating frequency, peak and basal pressure, and amplitude of voiding bladder contractions; immunohistochemical analysis for assessing the dopamine β-hydroxylase (DBH) and c-Fos levels; while body weight, head circumference, gait and dull general appearance were used for assessment of general condition od animals. The obtained results indicate that alterations in urinary function during hydrocephalus, are potentially linked to changes in descending noradrenergic modulation and diminished control of the vlPAG over the LC. The study is of sufficient significance and originality, presented results are quite convincing, however numerous issues need to be addressed.

QUESTION: Section Abstract should be rewritten since it is poorly fitted and confusing. For instance, the sentence: “The neuronal network controlling urination is complex and involves several brain areas, some of which are in circumventricular location, namely the periaqueductal gray (PAG) and the locus coeruleus (LC), and patients with hydrocephalus usually present urinary impairments.” is confusing. More important is that the presentation of results lacks clarity due to a conflation of the methods employed and the findings obtained.

ANSWER:

We acknowledge this remark. We rewrote the Abstract and separated the methods from the findings.

QUESTION: In section Introduction, lines 90-94, the sentence: “Since knowledge about the mechanism behind urinary dysfunction during hydrocephalus is scarce and considering that two key areas involved in the neuronal control of micturition are affected in this condition, namely the PAG and the LC, our aim was to determine the micturition profile of hydrocephalic animals and evaluate the degree of neuronal activation at the PAG [11], [12].” contains references. In the sentence where the objectives of the study are defined, explicit references are usually not provided. This section typically focuses on describing the problem, formulating research questions, and justifying the study's purpose. References are more commonly utilized in the literature review section, theoretical framework, or when citing previous research relevant to the establishment of objectives.

ANSWER: We acknowledge the importance of this remark. The references have been eliminated.

QUESTION: While for the sentence: “To answer those questions, we used a validated animal model, the rat injected with kaolin in the cisterna magna, used on our previous studies about noradrenergic pain modulation in hydrocephalic animals.” the reference(s) for the described animal model are lacking and should be provided.

ANSWER: We acknowledge the importance of this remark. The references for the described animal model have been added.

QUESTION: For sections Materials and methods, Results and Discussion, bellow are listed just some examples of the shortcomings of these sections and the study in general. It is necessary to complement the study with the required methods and results, as well as discussion.

  1. a)   Besides the investigated changes of the body weight, head circumference, gait and dull general appearance, the authors should incorporate testing of additional clinical signs and adequate behavior assessments, including neurological function, to strengthen the presented results.
  2. b)    Beyond alterations in the levels of the examined molecules, do their gene expressions, activities, cell locations, etc., also undergo changes? The information should be provided.
  3. c) Since the expression of DBH was assessed while DBH primarily contributes to catecholamine and trace amine biosynthesis, as well as participates in the metabolism of xenobiotics related to these substances, are the levels of those molecules also changed? Similarly, with regards to c-Fos, have there been alterations in the ascending/descending molecules?
  4. d)   It is necessary to ascertain whether gender disparities in the expression of the investigated parameters exist.

ANSWERS:

  1. We added some information regarding locomotor characterization of hydrocephalic animals. Please see lines 146-156 and 250-254.
  2. We did not measure gene expression but rather focus on the expression of the proteins. The reason for focusing on the proteins is that it has been shown that gene changes can occur without alterations in protein expression and due to the fact that this study is part of a characterization of mechanisms underlying dysfunctions in the hydrocephalic model and we previously measured protein expression. We now better justify the relation of the present study with previous studies (lines 103-104).
  3. We focused our analysis in the expression of DBH and c-Fos and not on ascending/descending molecules. The reasons to focus on these molecules is related to the use of methods that were already employed in our previous studies with the characterization of this animal model (Louçano et al., 2022), in order to compare the results of our different studies. Nevertheless, the published literature allows us to focus at the use of DBH as a marker of noradrenergic pathways and c-fos as marker of neuronal activation. In what concerns c-fos, several publications show that it is the most widely used functional anatomical marker of activated neurons in the central nervous system (Kovaks, 2008, Perrin-Terrin et al., 2016; Santos et al., 2018). Please see lines 214-220. As to the immunohistochemichal detection of DBH as an approach to identify noradrenergic neurons or fibers, since the seminal work of Hartman et al. 1972 and Hartman et al., 1972; Pickel and Reis 1976, the immunohistochemical detection of DBH is widely used as it is considered a specific noradrenergic marker with a perfect expression at noradrenergic fibers and is widely used to map noradrenergic innervation of the central nervous system (Cerpa et al., 2019) and allowing to quantify variations in innervation (Allard et al., 2011). We added this information to the Materials and Methods to better justify the use of immunostaing for DBH fibers (190-192).
  4. The question of gender disparities is challenging and we acknowledge the possibility of discussing these issues. In fact, we only used male rats and not females, as part of a project aiming to study hydrocephalus. Nevertheless, we should acknowledge that the results obtained in male rats cannot be directly translated to female animals. We added this to the Discussion. Please see lines 373-378.

QUESTION: Moreover, the comprehensive information about each used devices, chemicals, software, etc., should be provided (including the manufacturer, catalog number, etc.). The authors should uniform this item

ANSWER: We added all the available details, as requested.

QUESTION: The parts of subsections describing the immunohistochemical analyses of DHB and c-Fos are containing the same or similar information. It should be integrated.

ANSWER: We integrated the 2 subsections.

QUESTION: Detailed legends should be provided, including the explanations for used abbreviations.

ANSWER: The explanations for the abbreviations used in the legends of the figures have been added.

QUESTION: The punctuation marks and upper and lowercase letters should be used in appropriate places.

ANSWER: We acknowledge the importance of this remark. We have corrected the lowercase letters.

QUESTION: The same abbreviations and their explanations are introduced for several times in the manuscript, while several abbreviations are not defined. The authors should read the manuscript thoroughly and uniform this item.

ANSWER: We acknowledge the importance of this remark. The manuscript was reviewed, missing abbreviations were added, and repeated explanations were eliminated.

Reviewer 2 Report

Comments and Suggestions for Authors

For investigating the mechanisms underlying urinary dysfunction in a rat model with kaolin-induced hydrocephalus, the authors used cystometric analysis for evaluating frequency, peak and basal pressure, and amplitude of voiding bladder contractions; immunohistochemical analysis for assessing the dopamine β-hydroxylase (DBH) and c-Fos levels; while body weight, head circumference, gait and dull general appearance were used for assessment of general condition od animals. The obtained results indicate that alterations in urinary function during hydrocephalus, are potentially linked to changes in descending noradrenergic modulation and diminished control of the vlPAG over the LC. The study is of sufficient significance and originality, presented results are quite convincing, however numerous issues need to be addressed.

Concerns:

1. Section Abstract should be rewritten since it is poorly fitted and confusing. For instance, the sentence: “The neuronal network controlling urination is complex and involves several brain areas, some of which are in circumventricular location, namely the periaqueductal gray (PAG) and the locus coeruleus (LC), and patients with hydrocephalus usually present urinary impairments.” is confusing. More important is that the presentation of results lacks clarity due to a conflation of the methods employed and the findings obtained.

2. In section Introduction, lines 90-94, the sentence: “Since knowledge about the mechanism behind urinary dysfunction during hydrocephalus is scarce and considering that two key areas involved in the neuronal control of micturition are affected in this condition, namely the PAG and the LC, our aim was to determine the micturition profile of hydrocephalic animals and evaluate the degree of neuronal activation at the PAG [11], [12].” contains references. In the sentence where the objectives of the study are defined, explicit references are usually not provided. This section typically focuses on describing the problem, formulating research questions, and justifying the study's purpose. References are more commonly utilized in the literature review section, theoretical framework, or when citing previous research relevant to the establishment of objectives.

While for the sentence: “To answer those questions, we used a validated animal model, the rat injected with kaolin in the cisterna magna, used on our previous studies about noradrenergic pain modulation in hydrocephalic animals.” the reference(s) for the described animal model are lacking and should be provided.

3. For sections Materials and methods, Results and Discussion, bellow are listed just some examples of the shortcomings of these sections and the study in general. It is necessary to complement the study with the required methods and results, as well as discussion.

a)   Besides the investigated changes of the body weight, head circumference, gait and dull general appearance, the authors should incorporate testing of additional clinical signs and adequate behavior assessments, including neurological function, to strengthen the presented results.

b)      Beyond alterations in the levels of the examined molecules, do their gene expressions, activities, cell locations, etc., also undergo changes? The information should be provided.

c) Since the expression of DBH was assessed while DBH primarily contributes to catecholamine and trace amine biosynthesis, as well as participates in the metabolism of xenobiotics related to these substances, are the levels of those molecules also changed? Similarly, with regards to c-Fos, have there been alterations in the ascending/descending molecules?

d)   It is necessary to ascertain whether gender disparities in the expression of the investigated parameters exist.

Moreover, the comprehensive information about each used devices, chemicals, software, etc., should be provided (including the manufacturer, catalog number, etc.). The authors should uniform this item.

The parts of subsections describing the immunohistochemical analyses of DHB and c-Fos are containing the same or similar information. It should be integrated.

4.  Detailed legends should be provided, including the explanations for used abbreviations.

5.  The punctuation marks and upper and lowercase letters should be used in appropriate places.

6. The same abbreviations and their explanations are introduced for several times in the manuscript, while several abbreviations are not defined. The authors should read the manuscript thoroughly and uniform this item.

Comments on the Quality of English Language

Minor editing of English language required

Author Response

Response to reviewers- Manuscript ID: biomedicines-2797923 “Noradrenergic pathways involved in micturition in an animal model of hydrocephalus – implications for urinary dysfunction”

Reviewer 1

The author creatively discusses the neural mechanism of urinary function impairment in patients with hydrocephalus, but the discussion of the mechanism can be further deepened. No matter the expression of DBH at the Onuf’s Nucleus or the decreasing in neuronal activation of the vlPAG in hydrocephalic animals, the author only made a symbolic observation and did not explore the mechanism. In general, the author needs to increase the necessary mechanism to increase the scientific significance of the article.

ANSWER: We agree with this remark and changed the manuscript accordingly to propose a mechanism based on the current and previous findings. We changed the abstract to summarize the mechanism proposed (please see lines 35-39) and the Discussion (please see the paragraph that starts in line 366).

Reviewer 2

For investigating the mechanisms underlying urinary dysfunction in a rat model with kaolin-induced hydrocephalus, the authors used cystometric analysis for evaluating frequency, peak and basal pressure, and amplitude of voiding bladder contractions; immunohistochemical analysis for assessing the dopamine β-hydroxylase (DBH) and c-Fos levels; while body weight, head circumference, gait and dull general appearance were used for assessment of general condition od animals. The obtained results indicate that alterations in urinary function during hydrocephalus, are potentially linked to changes in descending noradrenergic modulation and diminished control of the vlPAG over the LC. The study is of sufficient significance and originality, presented results are quite convincing, however numerous issues need to be addressed.

QUESTION: Section Abstract should be rewritten since it is poorly fitted and confusing. For instance, the sentence: “The neuronal network controlling urination is complex and involves several brain areas, some of which are in circumventricular location, namely the periaqueductal gray (PAG) and the locus coeruleus (LC), and patients with hydrocephalus usually present urinary impairments.” is confusing. More important is that the presentation of results lacks clarity due to a conflation of the methods employed and the findings obtained.

ANSWER:

We acknowledge this remark. We rewrote the Abstract and separated the methods from the findings.

QUESTION: In section Introduction, lines 90-94, the sentence: “Since knowledge about the mechanism behind urinary dysfunction during hydrocephalus is scarce and considering that two key areas involved in the neuronal control of micturition are affected in this condition, namely the PAG and the LC, our aim was to determine the micturition profile of hydrocephalic animals and evaluate the degree of neuronal activation at the PAG [11], [12].” contains references. In the sentence where the objectives of the study are defined, explicit references are usually not provided. This section typically focuses on describing the problem, formulating research questions, and justifying the study's purpose. References are more commonly utilized in the literature review section, theoretical framework, or when citing previous research relevant to the establishment of objectives.

ANSWER: We acknowledge the importance of this remark. The references have been eliminated.

QUESTION: While for the sentence: “To answer those questions, we used a validated animal model, the rat injected with kaolin in the cisterna magna, used on our previous studies about noradrenergic pain modulation in hydrocephalic animals.” the reference(s) for the described animal model are lacking and should be provided.

ANSWER: We acknowledge the importance of this remark. The references for the described animal model have been added.

QUESTION: For sections Materials and methods, Results and Discussion, bellow are listed just some examples of the shortcomings of these sections and the study in general. It is necessary to complement the study with the required methods and results, as well as discussion.

a)   Besides the investigated changes of the body weight, head circumference, gait and dull general appearance, the authors should incorporate testing of additional clinical signs and adequate behavior assessments, including neurological function, to strengthen the presented results.

b)    Beyond alterations in the levels of the examined molecules, do their gene expressions, activities, cell locations, etc., also undergo changes? The information should be provided.

c) Since the expression of DBH was assessed while DBH primarily contributes to catecholamine and trace amine biosynthesis, as well as participates in the metabolism of xenobiotics related to these substances, are the levels of those molecules also changed? Similarly, with regards to c-Fos, have there been alterations in the ascending/descending molecules?

d)   It is necessary to ascertain whether gender disparities in the expression of the investigated parameters exist.

ANSWERS:

a) We added some information regarding locomotor characterization of hydrocephalic animals. Please see lines 146-156 and 250-254.

b) We did not measure gene expression but rather focus on the expression of the proteins. The reason for focusing on the proteins is that it has been shown that gene changes can occur without alterations in protein expression and due to the fact that this study is part of a characterization of mechanisms underlying dysfunctions in the hydrocephalic model and we previously measured protein expression. We now better justify the relation of the present study with previous studies (lines 103-104).

c) We focused our analysis in the expression of DBH and c-Fos and not on ascending/descending molecules. The reasons to focus on these molecules is related to the use of methods that were already employed in our previous studies with the characterization of this animal model (Louçano et al., 2022), in order to compare the results of our different studies. Nevertheless, the published literature allows us to focus at the use of DBH as a marker of noradrenergic pathways and c-fos as marker of neuronal activation. In what concerns c-fos, several publications show that it is the most widely used functional anatomical marker of activated neurons in the central nervous system (Kovaks, 2008, Perrin-Terrin et al., 2016; Santos et al., 2018). Please see lines 214-220. As to the immunohistochemichal detection of DBH as an approach to identify noradrenergic neurons or fibers, since the seminal work of Hartman et al. 1972 and Hartman et al., 1972; Pickel and Reis 1976, the immunohistochemical detection of DBH is widely used as it is considered a specific noradrenergic marker with a perfect expression at noradrenergic fibers and is widely used to map noradrenergic innervation of the central nervous system (Cerpa et al., 2019) and allowing to quantify variations in innervation (Allard et al., 2011). We added this information to the Materials and Methods to better justify the use of immunostaing for DBH fibers (190-192).

d) The question of gender disparities is challenging and we acknowledge the possibility of discussing these issues. In fact, we only used male rats and not females, as part of a project aiming to study hydrocephalus. Nevertheless, we should acknowledge that the results obtained in male rats cannot be directly translated to female animals. We added this to the Discussion. Please see lines 373-378.

QUESTION: Moreover, the comprehensive information about each used devices, chemicals, software, etc., should be provided (including the manufacturer, catalog number, etc.). The authors should uniform this item

ANSWER: We added all the available details, as requested.

QUESTION: The parts of subsections describing the immunohistochemical analyses of DHB and c-Fos are containing the same or similar information. It should be integrated.

ANSWER: We integrated the 2 subsections.

QUESTION: Detailed legends should be provided, including the explanations for used abbreviations.

ANSWER: The explanations for the abbreviations used in the legends of the figures have been added.

QUESTION: The punctuation marks and upper and lowercase letters should be used in appropriate places.

ANSWER: We acknowledge the importance of this remark. We have corrected the lowercase letters.

QUESTION: The same abbreviations and their explanations are introduced for several times in the manuscript, while several abbreviations are not defined. The authors should read the manuscript thoroughly and uniform this item.

ANSWER: We acknowledge the importance of this remark. The manuscript was reviewed, missing abbreviations were added, and repeated explanations were eliminated.

Round 2

Reviewer 1 Report

Comments and Suggestions for Authors

well revised

Reviewer 2 Report

Comments and Suggestions for Authors

In the revised version of the manuscript, the authors have addressed the majority of my comments and concerns that were raised on their original submission. The overall quality of the manuscript has been significantly improved. I find the article as substantial contribution in its field while the requirements for a publication in Biomedicine are fulfilled. Therefore, I recommend publication.